# Overexpression of Neurogenin 1 Negatively Regulates Osteoclast and Osteoblast Differentiation

**DOI:** 10.3390/ijms23126708

**Published:** 2022-06-16

**Authors:** Jung Ha Kim, Kabsun Kim, Inyoung Kim, Semun Seong, Jeong-Tae Koh, Nacksung Kim

**Affiliations:** 1Department of Pharmacology, Chonnam National University Medical School, Gwangju 61469, Korea; kjhpw@hanmail.net (J.H.K.); kabsun@hanmail.net (K.K.); doll517@naver.com (I.K.); iamsemun@chonnam.edu (S.S.); 2Hard-Tissue Biointerface Research Center, School of Dentistry, Chonnam National University, Gwangju 61186, Korea; jtkoh@chonnam.ac.kr; 3Department of Pharmacology and Dental Therapeutics, School of Dentistry, Chonnam National University, Gwangju 61186, Korea

**Keywords:** Neurogenin 1, osteoclast, osteoblast, PCAF, ankylosing spondylitis

## Abstract

Neurogenin 1 (Ngn1) belongs to the basic helix–loop–helix (bHLH) transcription factor family and plays important roles in specifying neuronal differentiation. The present study aimed to determine whether forced Ngn1 expression contributes to bone homeostasis. Ngn1 inhibited the p300/CREB-binding protein-associated factor (PCAF)-induced acetylation of nuclear factor of activated T cells 1 (NFATc1) and runt-related transcription factor 2 (Runx2) through binding to PCAF, which led to the inhibition of osteoclast and osteoblast differentiation, respectively. In addition, Ngn1 overexpression inhibited the TNF-α- and IL-17A-mediated enhancement of osteoclast differentiation and IL-17A-induced osteoblast differentiation. These findings indicate that Ngn1 can serve as a novel therapeutic agent for treating ankylosing spondylitis with abnormally increased bone formation and resorption.

## 1. Introduction

Bone homeostasis is maintained by the balance of tightly coupled processes between bone resorption by osteoclasts and bone formation by osteoblasts. However, an imbalance in these processes leads to skeletal disorders, such as osteoporosis, osteopetrosis, and rheumatoid arthritis [1,2,3]. Osteoclasts are giant multinucleate cells (with diameters of up to 100 µm) that are differentiated from bone marrow hematopoietic precursor cells under the control of macrophage colony stimulating factor (M-CSF) and the receptor activator of nuclear factor kappa-B ligand (RANKL). RANKL and the receptor activator of nuclear factor kappa-B (RANK) constitute the key signaling pathway for osteoclast differentiation and function. This pathway induces the expression of several transcription factors, including the master regulator of osteoclastogenesis, the nuclear factor of activated T cells 1 (NFATc1), and osteoclast-specific genes, such as tartrate-resistant acid phosphatase (TRAP, *Acp5*), the osteoclast-associated receptor (*Oscar*), and cathepsin K (*Ctsk*) [4,5,6,7]. Osteoblasts are differentiated from mesenchymal stem cells (MSCs) under the control of multiple signaling pathways, such as the bone morphogenetic protein (BMP), Wnt, and notch signaling pathways [8]. All these pathways play important roles in the expression and activity of the master transcription factor of osteoblastogenesis, namely, runt-related transcription factor 2 (Runx2) [8,9]. Runx2 induces the commitment of MSCs to the osteogenic lineage and stimulates osteoblast differentiation by inducing the expression of osteoblast-specific genes, such as alkaline phosphatase (ALP, *Alpl*), osterix (*Sp7*), bone sialoprotein (BSP, *Ibsp*), and osteocalcin (*Bglap*) by binding to the osteoblast-specific acting element (OSE) present in the promoter region [10,11,12].

Neurogenin 1 (Ngn1) belongs to the family of basic helix–loop–helix (bHLH) transcription factors. *Ngn1*-null mouse models have revealed that the gene plays a critical role in the differentiation of glutamatergic spiral ganglion neurons during development. It contributes to the normal differentiation of glutamatergic spiral ganglion neurons by activating the downstream cascade of NeuroD1, Brm3a, GATA3, and neurotrophic factor receptors. Furthermore, in vitro studies using an Ngn1 overexpression model have revealed that Ngn1 stimulates the neuronal differentiation of neural progenitors but inhibits glial differentiation [13,14,15,16].

The neural and skeletal systems are physically and functionally associated [17,18,19,20,21]. Impaired innervation in skeletal disorders, such as osteoporosis, osteoarthritis, and neurogenic heterotopic ossification, indicate a close interaction between skeletal and neural systems. Moreover, the crosstalk between osteoclasts or osteoblasts and neurons is supported by both in vivo and in vitro studies. These studies have demonstrated that correct innervation is important to maintain bone homeostasis, skeletal growth, and fracture repair. In addition, a disrupted skeletal system affects nerve formation and signaling [17]. Moreover, neural and skeletal systems share various molecules and regulatory mechanisms, such as BMPs, Eph-ephrin, netrins, semaphorins, and Slit/Robo signaling [17,20,22,23,24,25,26,27,28].

The present study aimed to determine whether Ngn1 affects osteoclast and osteoblast differentiation. We found that the overexpression of Ngn1 in osteoclast or osteoblast precursor cells negatively regulated osteoclast or osteoblast differentiation by inhibiting the transcriptional activity of NFATc1 or Runx2, the master regulators of differentiation into osteoclasts or osteoblasts, respectively. The regulation of these two transcription factors by Ngn1 was associated with a direct interaction between Ngn1 and p300/CREB-binding protein-associated factor (PCAF).

## 2. Results

### 2.1. Ngn1 Inhibited Osteoclast and Osteoblast Differentiation

Since *Ngn1* expression is very low during osteoclastogenesis and osteoblastogenesis, we overexpressed Ngn1 in osteoclast and osteoblast precursor cells using a retrovirus to investigate its role in bone cells. To determine the role of Ng1 in osteoclast differentiation, Ngn1 was overexpressed in BMMs. Ngn1 overexpression significantly suppressed the formation of TRAP-positive osteoclasts (Figure 1a) and significantly downregulated the expression of osteoclastogenic marker genes, such as *Nfatc1*, *Acp5*, *Oscar*, and *Ctsk* (Figure 1b).

Ngn1 was then overexpressed in osteoblast precursors to evaluate the effect of Ngn1 on osteoblast differentiation. Ngn1 overexpression in osteoblast precursors resulted in the remarkable inhibition of ALP activity and mineralization (Figure 2a,b). In addition, Ngn1 overexpression significantly suppressed the expression of osteoblastogenic marker genes, such as *Runx2*, *Alpl*, *Ibsp*, and *Bglap*, during osteoblast differentiation (Figure 2c). Collectively, the overexpression of Ngn1 inhibits the differentiation of the two major types of bone cells.

### 2.2. Ngn1 Inhibited NFATc1 and Runx2 Transcriptional Activities through Interaction with PCAF

Since the overexpression of Ngn1 inhibited the expression of differentiation marker genes in both osteoclasts and osteoblasts, we speculated that Ngn1 might inhibit the transcriptional activities of master transcription factors of osteoclast or osteoblast differentiation through the regulation of certain activators that are simultaneously involved in both osteoclast and osteoblast differentiation, independently of their DNA-binding ability. PCAF is a coactivator that regulates both osteoclast and osteoblast differentiation. To determine whether Ngn1 sequesters PCAF to inhibit NFATc1 or Runx2 target gene expression, we investigated whether Ngn1 directly binds to PCAF. When Ngn1 and PCAF were co-transfected and immunoprecipitated, a physical interaction was observed between Ngn1 and PCAF (Figure 3a). We then investigated the role of Ngn1 in the PCAF-mediated acetylation of NFATc1 and Runx2 using immunoprecipitation assays. As shown in Figure 3b and c, Ngn1 remarkably suppressed the acetylation of NFATc1 and Runx2 mediated by PCAF. Upon observing repressed acetylation, we subsequently explored the role of Ngn1 in the transcriptional activity of NFATc1 driven by *Acp5* and *Oscar* promoters using luciferase assays. The results show that Ngn1 significantly suppressed the NFATc1-induced and PCAF-enhanced transcription of NFATc1 target genes, namely, *Acp5* and *Oscar* (Figure 3d). Furthermore, the effects of Ngn1 on the transcriptional activity of Runx2 driven by *Alpl*, *Bglap*, and *OSE* downstream of Runx2 were determined using luciferase assays. Ngn1 significantly suppressed the transcription of Runx2 downstream targets induced by PCAF (Figure 3e). These results suggested that the interaction between Ngn1 and PCAF suppressed the NFATc1 and Runx2 acetylation mediated by PCAF, thereby suppressing the expression of their target genes.

We then examined whether NFATc1 or Runx2 could rescue defects in osteoclastogenesis or osteoblastogenesis, respectively, caused by Ngn1 overexpression. The overexpression of the constitutively active form of NFATc1 (Ca-NFATc1) partially but significantly rescued the Ngn1-induced suppression of osteoclast formation (Figure 4a). Furthermore, the overexpression of Runx2 partially but significantly rescued the inhibitory effects of Ngn1 during osteoblastogenesis (Figure 4b,c). Collectively, these results suggest that the simultaneous inhibition of osteoclast and osteoblast differentiation by Ngn1 is associated, at least in part, with reduced PCAF-mediated NFATc1 and Runx2 acetylation.

### 2.3. Ngn1 Inhibited the Osteoclast and Osteoblast Differentiation Mediated by Inflammatory Cytokines

Osteoclast differentiation, although mainly induced by RANKL, can also be induced by other inflammatory cytokines [29,30]. Therefore, we analyzed whether Ngn1 can inhibit the osteoclast formation induced by inflammatory cytokines, such as TNF-α and IL-17A. Ngn1 overexpression completely inhibited the osteoclast formation induced by TNF-α (Figure 5a) and IL-17A (Figure 5b), as observed via TRAP staining. Furthermore, as IL-17A is known to induce osteoblast differentiation, we investigated the effects of Ngn1 on IL-17A-induced osteoblast differentiation. Ngn1 overexpression significantly inhibited ALP activity, as well as the mineralization induced by IL-17A (Figure 5c,d). Collectively, Ngn1 inhibited RANKL-induced osteoclast and BMP2-induced osteoblast differentiation, as well as the osteoclast and osteoblast differentiation induced by inflammatory cytokines, such as TNF-α and IL-17A.

## 3. Discussion

Several studies have shown that histone acetyltransferases, including PCAF, p300, monocytic leukemia zinc finger protein (MOZ), and MOZ-related factor (MORF), stimulate osteoclastogenesis and osteoblastogenesis through the acetylation of both histone and non-histone proteins [31]. PCAF is a histone acetyltransferase involved in tumor initiation and progression primarily via the acetylation of H3 histones. PCAF is also associated with multiple hepatic metabolic and pathogenic diseases, such as metabolic syndrome, inflammation, apoptosis, injury, and cancer, via the acetylation of non-histone proteins such as phosphoglycerate kinase 1 (PGK1), ATP-citrate lyase (ACLY), peroxisome proliferator-activated receptor gamma coactivator 1-alpha (PGC1-α), forkhead box P3 (FOXP3), and p53 [32]. Additionally, PCAF is also involved in the differentiation of bone cells via the acetylation of both histone and non-histone proteins.

Both NFATc1 and Runx2 are master transcription factors of osteoclast and osteoblast differentiation, respectively [33,34]. They are tightly regulated at the transcriptional, translational, and post-translational levels. Their stability and activities are affected by PCAF-mediated acetylation [35,36,37,38]. PCAF synergistically enhances RANKL-induced osteoclast differentiation by promoting the stability and transcriptional activity of NFATc1 through direct interaction [37]. Furthermore, PCAF binds to Runx2, thereby acetylating it and consequently increasing its stability and transcriptional activity. In addition, PCAF stimulates osteoblast differentiation through inducing Runx2 acetylation in MC3T3-E1 cells [39]. Therefore, PCAF positively regulates the differentiation of osteoclasts and osteoblasts by promoting the stability and activity of their respective master transcription factors.

Ngn1 regulates Ngn1-dependent transcription by binding to E-box elements as a bHLH transcription factor, and also acts as a transcription repressor independently of its ability to bind to DNA. For example, Ngn1 sequesters the transcriptional coactivator complex comprising the CREB binding protein (CBP)/p300 and Smad1 from STAT by directly interacting with CBP, thus inhibiting glial cell differentiation [40]. We found that Ngn1 overexpression simultaneously inhibited osteoclast and osteoblast differentiation and suppressed the expression of all their specific genes. These results indicate that Ngn1 simultaneously prevents the function of an activator of NFATc1 and Runx2 via protein–protein interactions rather than the transcriptional regulation of each gene through DNA binding. One suitable candidate that mediates the inhibitory function of Ngn1 on NFATc1 and Runx2 is PCAF, as it co-regulates osteoclast and osteoblast differentiation. Therefore, we investigated whether Ngn1 interacts with PCAF. The co-immunoprecipitation results reveal an interaction between Ngn1 and PCAF. Ngn1, when bound to PCAF, inhibited PCAF-mediated acetylation and the transcriptional activities of NFATc1 and Runx2. However, neither the overexpression of NFATc1 nor Runx2 rescued the inhibitory effects of Ngn1 completely. Therefore, Ngn1 may also regulate target gene expression depending on its ability to bind to DNA or other PCAF-dependent transcription factors to inhibit osteoclast and osteoblast differentiation.

Ankylosing spondylitis (AS) is a common inflammatory autoimmune disease that is involved in disorders of the immune and skeletal systems [41]. It is characterized by inflammatory damage to the axial skeleton and bony ankyloses [41]. The pathogenesis of AS is associated with infection and environmental and genetic factors [42,43]. This disease uncouples the processes between osteoclasts and osteoblasts, concomitantly resulting in osteogenesis and osteolytic bone destruction. Pathological changes in the spine during AS are characterized by an increased synthesis of new bone at sites of inflammation and excessive resorption of trabecular bone, with an increased number of osteoclasts [44,45,46,47,48,49]. Although TNF-α blockade strategies for AS have been developed, more effective and safer strategies are needed owing to the controversy over the effects of TNF-α on bone formation [50,51,52]. Patients with AS express abundant IL-17A, which is involved in rapid differentiation into mature osteoblasts and the promotion of osteoclast differentiation and resorption [50,53,54]. Hence, strategies to block IL-17A can be applied to treat AS.

In the present study, the overexpression of Ngn1 negatively regulated osteoblast differentiation. Ngn1 inhibited the osteoclast differentiation induced by RANKL, as well as by TNF-α and IL-17A. Furthermore, Ngn1 suppressed the osteogenic differentiation and function induced by BMP2 and IL-17A. Together, these results suggest that the administration of Ngn1 can ameliorate excessive bone formation and resorption in inflammatory sites of AS. While multiple in vitro studies suggest that Ngn1 overexpression promotes proliferation during the development of spiral ganglion neurons and neuronal differentiation in pluripotent stem cells, little is known about the function of Ngn1 overexpression in vivo, except for reports that retina-like tissue is induced by ectopic Ngn1 expression from the Bestrophin1 promoter [55,56]. Therefore, the therapeutic potential of Ngn1 in AS needs to be confirmed using bone cell-specific Ngn1-overexpressing mice. Nevertheless, due to a lack of information, it is difficult to predict the effects of Ngn1 overexpression in vivo.

## 4. Materials and Methods

### 4.1. Osteoclast Differentiation

Bonemarrow-derived macrophages (BMMs), isolated from the femurs and tibias of wild-type ICR mice by flushing with α-MEM (HyClone Laboratories, Logan, UT, USA), were cultured in α-MEM containing 10% FBS (HyClone Laboratories, Logan, UT, USA), 100 U/mL penicillin, 100 mg/mL streptomycin (Life Technologies, Carlsbad, CA, USA), and 30 ng/mL M-CSF, for 3 days. Nonadherent cells were removed, and adherent BMMs were differentiated into osteoclasts via incubation with α-MEM containing 10% FBS (HyClone Laboratories, Logan, UT, USA), 100 U/mL penicillin, 100 mg/mL streptomycin (Life Technologies, Carlsbad, CA, USA), M-CSF (30 ng/mL), and RANKL (10–150 ng/mL). Mature osteoclasts were fixed and stained for TRAP. Thereafter, TRAP-positive multinuclear cells with >3 nuclei were considered as osteoclasts.

### 4.2. Osteoblast Differentiation

Primary osteoblast precursors were isolated from the calvarias of neonatal ICR mice via enzymatic digestion with 0.1% collagenase (Life Technologies, Carlsbad, CA, USA) and 0.2% dispase II (Roche Diagnostics, GmbH, Mannheim, Germany). Primary osteoblast precursors were differentiated into osteoblasts by culturing in α-MEM containing 10% FBS (HyClone Laboratories, Logan, UT, USA), 100 U/mL penicillin, 100 mg/mL streptomycin (Life Technologies, Carlsbad, CA, USA), BMP2 (100 ng/mL), ascorbic acid (50 µg/mL), and β-glycerophosphate (100 mM). The cultured cells were lysed in 50 mM Tris-HCl (pH 7.4) containing 1% Triton X-100, 150 mM NaCl, and 1 mM EDTA and incubated with p-nitrophenyl phosphate substrate (Sigma-Aldrich, St. Louis, MO, USA). ALP activity was then assessed by measuring the absorbance at 405 nm using a spectrophotometer. Cultured cells were fixed with 4% paraformaldehyde, stained with 40 mM alizarin red (pH 4.2), and washed with phosphate-buffered saline (PBS) to remove nonspecific staining. Then, the cells were visualized using the CanoScan 9000F Mark II scanner (Canon Inc., Tokyo, Japan). The quantitation of alizarin red was performed by extracting alizarin red from the stained cells with 10% acetic acid for 30 min. Subsequently, the absorbance of the samples was measured at 405 nm via spectrophotometry.

### 4.3. Retroviral Transduction

Plat-E cells were transfected using FuGENE 6 (Promega, Madison, WI, USA), as described by the manufacturer, to produce retroviral packages. Retroviral supernatants were collected 48 h post-transfection and then incubated with the cells of interest for 6 h with 10 μg/mL polybrene (Sigma-Aldrich Corp., St. Louis, MO, USA).

### 4.4. Quantitative Reverse-Transcription PCR (qRT-PCR)

We amplified the sequences of interest using specific primers in triplicate via qRT-PCR using SYBR Green (Qiagen, GmbH, Hilden, Germany) and Rotor-Gene Q (Qiagen). The transcript-level expression of target genes was normalized to that of *Gapdh*. The relative quantified value for the expression of each target gene compared with its calibrator is expressed as 2^−(Ct−Cc)^, where ^Ct^ and ^Cc^ are the mean threshold cycle differences of the target and calibrator genes, respectively, after normalization to *Gapdh* expression. The relative expression for each sample is shown in a semi-log plot. The forward and reverse primer sequences (5′ → 3′) were as follows: *Gapdh*, TGACCACAGTCCATGCCATCACTG and CAGGAGACAACCTGGTCCTCAGTG; *Nfatc1*, CTCGAAAGACAGCACTGGAGCAT and CGGCTGCCTTCCGTCTCATAG; *Acp5*, CTGGAGTGCACGATGCCAGCGACA and TCCGTGCTCGGCGATGGACCAGA; *Oscar*, TGCTGGTAACGGATCAGCTCCCCAGA and CCAGGAGCCAGAACCTTCGAAACT; *Ctsk*, AGGAGGCATTGACTCTGAGATG and GTTGTTCTTATCCGAGCCAAGAG; *Runx2*, CCCAGCCACCTTTACCTACA and CAGCGTCAACACCATCATTC; *Alpl*, CAAGGATATCGACGTGATCATG and GTCAGTCAGGTTGTTCCGATTC; *Ibsp*, GGAAGGGAGACTTCAAACGAAG and CATCCACTTCTGCTTCTTCGT TC; and *Bglap*, ATGAGGACCCTCTCTCTGCTGCTCAC and AGAGCAAACTGCAGAAGCTGAGAG.

### 4.5. Luciferase Reporter Assay

We transfected 293T cells with *Acp5*, *Oscar*, *Alpl*, *Bglap*, or *OSE* reporter plasmids using FuGENE 6 for 48 h (Promega, Madison, WI, USA), as described by the manufacturer. The cells were lysed in Passive Lysis Buffer (Promega, Madison, WI, USA), and luciferase activity was measured in duplicate using the dual-luciferase reporter assay system (Promega, Madison, WI, USA).

### 4.6. Immunoprecipitation

We transfected 293T cells with the peptide sequence DYKDDDDK (Flag)-PCAF, Flag-Neurogenin 1, human influenza agglutinin-NFATc1 (HA-NFATc1), or anti-master regulator of cell cycle entry and proliferative metabolism-Runx2 (Myc-Runx2), as indicated. The cells were washed with PBS and lysed with 50 mM Tris-HCl (pH 8.0), 150 mM NaCl, 1 mM EDTA, 0.5% Nonidet P-40, 1 mM PMSF, and a protease inhibitor cocktail. Proteins in the lysates were immunoprecipitated using PCAF (Santa Cruz Biotechnology, Dallas, TX, USA) or Ac-lysine (Cell Signaling Technology, Danvers, MA, USA) antibodies overnight at 4 °C. Protein–antibody complexes were extracted using Pierce™ Protein G Agarose (Thermo Fisher Scientific, Waltham, MA, USA); then, the proteins were separated via SDS-PAGE and transferred onto PVDF membranes (Millipore, Burlington, MA, USA). Nonspecific binding on the membranes was blocked by treating them with 5% skim milk in 10 mM Tri-HCl (pH 7.6), 150 mM NaCl, and 0.1% Tween 20 (TBS-T). The membranes were then immunoblotted with anti-Flag antibodies (Sigma-Aldrich Corp., St. Louis, MO, USA), anti-NFATc1 antibodies (Santa Cruz Biotechnology, Dallas, TX, USA), or anti-–Myc antibodies (Santa Cruz Biotechnology, Dallas, TX, USA). Signals were detected using ECL solution (Millipore, Burlington, MA, USA) and analyzed using the Azure 300 luminescent image analyzer (Azure Biosystems, Dublin, CA, USA).

### 4.7. Statistical Analysis

All values are expressed as mean ± standard deviation (SD). Statistical significance was determined using a two-tailed Student’s *t*-test for two independent samples or analysis of variance (ANOVA) with post hoc Tukey HSD test for multiple group comparisons. Results with *p* < 0.05 were considered statistically significant.

## Figures and Tables

**Figure 1 ijms-23-06708-f001:**
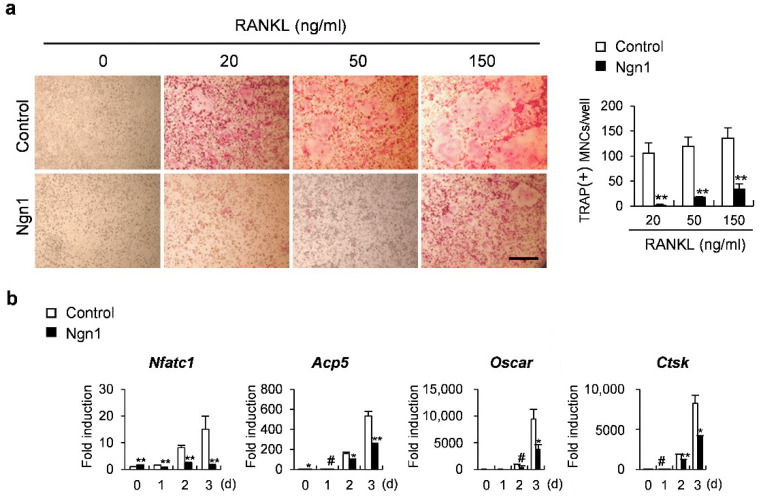
The overexpression of Neurogenin 1 (Ngn1) inhibited the receptor activator of nuclear factor kappa-B ligand (RANKL)-induced osteoclast differentiation. (**a**,**b**) Bone marrow-derived macrophages (BMMs) overexpressing Ngn1 were generated via retroviral infection and cultured with a macrophage colony stimulating factor (M-CSF) and RANKL for 3 days. (**a**) Tartrate-resistant acid phosphatase (TRAP)-stained cells (left panel) and the number of TRAP-positive multinucleated cells (right panel), scale bar: 200 µm. (**b**) Relative expression of the indicated genes quantified using quantitative reverse-transcription PCR (qRT-PCR). # *p* < 0.05, * *p* < 0.01, ** *p* < 0.001 vs. control.

**Figure 2 ijms-23-06708-f002:**
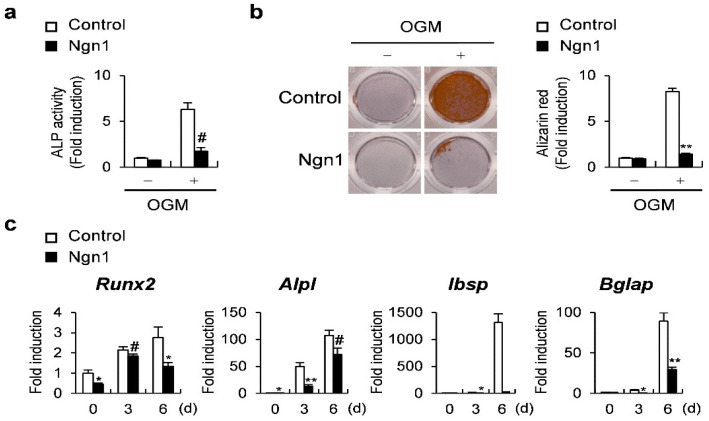
The overexpression of Ngn1 strongly inhibited osteoblast differentiation. (**a**–**c**) Osteoblast precursor cells overexpressing Ngn1 were generated via retroviral infection and incubated in an osteogenic medium (OGM). (**a**) Levels of alkaline phosphatase (ALP) in cells cultured for 3 days. (**b**) The cells were cultured for 6 days, stained with alizarin red (left panel), and those positive for alizarin red were quantified (right panel). (**c**) Relative expression of the indicated genes quantified via quantitative reverse-transcription PCR (qRT-PCR). # *p* < 0.05, * *p* < 0.01, ** *p* < 0.001 vs. control.

**Figure 3 ijms-23-06708-f003:**
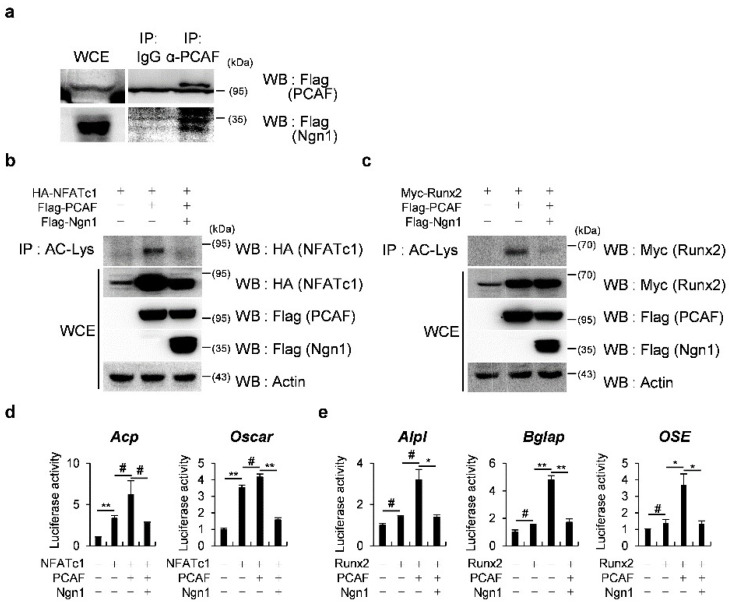
Ngn1 regulated the p300/CREB-binding protein-associated factor (PCAF)-mediated transcriptional activities of the nuclear factor of activated T cells 1 (NFATc1) and runt-related transcription factor 2 (Runx2). (**a**) The 293T cells were co-transfected with Flag-PCAF and Flag-Ngn1 and subjected to immunoprecipitation with an IgG or anti-PCAF antibody, followed by Western blot analysis with an anti-Flag antibody. (**b**) The 293T cells were co-transfected with HA-NFATc1, Flag-PCAF, or Flag-Ngn1, as indicated. Cell lysates were subjected to immunoprecipitation with an anti-acetyl-lysine antibody, followed by Western blot analysis of the indicated antibodies. (**c**) The 293T cells were co-transfected with Myc-Runx2, Flag-PCAF, or Flag-Ngn1, as indicated. Cell lysates were subjected to immunoprecipitation with an anti-acetyl-lysine antibody, followed by Western blot analysis with the indicated antibodies. (**d**) The 293T cells were co-transfected with the indicated plasmids along with an *Acp5* or *Oscar* promoter luciferase reporter. Cell lysates were subjected to the luciferase assay. (**e**) The 293T cells were co-transfected with the indicated plasmids along with an *Alpl*, *Bglap*, or *OSE* promoter luciferase reporter. Cell lysates were subjected to the luciferase assay. # *p* < 0.05, * *p* < 0.01, ** *p* < 0.001 vs. control.

**Figure 4 ijms-23-06708-f004:**
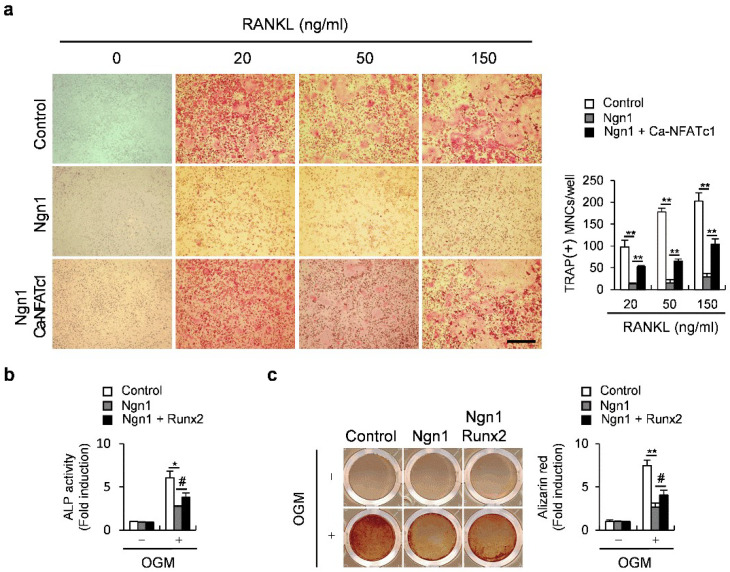
The overexpression of NFATc1 or Runx2 partially rescued the inhibitory effects of Ngn1 on osteoclastogenesis or osteoblastogenesis, respectively. (**a**) BMMs overexpressing Ngn1 or Ngn1 and Ca-NFATc1 were generated via retroviral infection and subsequently cultured with M-CSF and RANKL for 3 days. TRAP-stained cells (left panel) and the number of TRAP-positive multinucleated cells (right panel), scale bar: 200 µm. (**b**,**c**) Osteoblast precursor cells overexpressing Ngn1 or Ngn1 and Runx2 were generated via retroviral infection and subsequently cultured with OGM. (**b**) Levels of ALP in cells cultured for 3 days. (**c**) Cells were cultured for 6 days, stained with alizarin red (left panel), and those positive for alizarin red were quantified (right panel). # *p* < 0.05, * *p* < 0.01, ** *p* < 0.001 vs. control.

**Figure 5 ijms-23-06708-f005:**
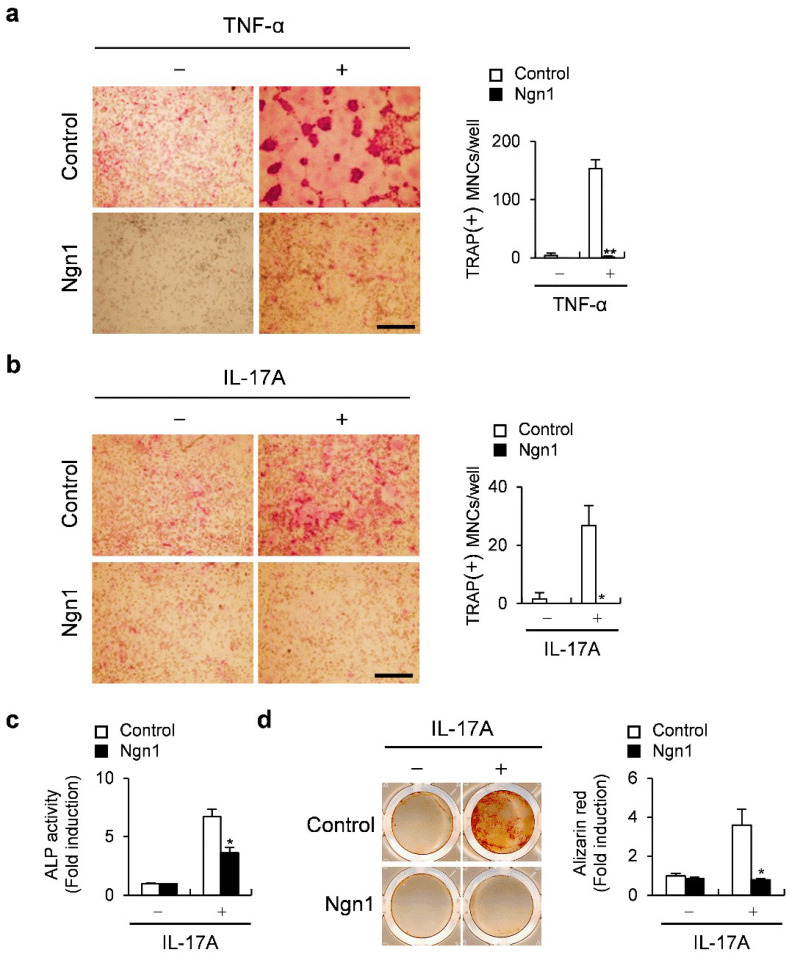
The overexpression of Ngn1 inhibited the osteoclastogenesis and osteoblastogenesis mediated by inflammatory cytokines. (**a**) BMMs overexpressing Ngn1 were generated via retroviral infection and subsequently incubated with M-CSF, RANKL, and TNF-α for 3 days. TRAP-stained cells (left panel) and the number of TRAP-positive multinucleated cells (right panel), scale bar: 200 µm. (**b**) BMMs overexpressing Ngn1 were generated via retroviral infection and subsequently incubated with M-CSF, RANKL, and IL-17A for 3 days. TRAP-stained cells (left panel) and the number of TRAP-positive multinucleated cells (right panel), scale bar: 200 µm. (**c**,**d**) Osteoblast precursor cells overexpressing Ngn1 were generated via retroviral infection and subsequently cultured with ascorbic acid, β-glycerophosphate, and IL-17A. (**c**) Levels of ALP in cells cultured for 3 days. (**d**) Cells were cultured for 6 days, stained with alizarin red (left panel), and those positive for alizarin red were quantified (right panel). * *p* < 0.01, ** *p* < 0.001 vs. control.

## Data Availability

All data generated or analyzed during this study are included in this published article.

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
