# Peer review of "Overexpression of Neurogenin 1 Negatively Regulates Osteoclast and Osteoblast Differentiation"

_ijms, 2022, doi:10.3390/ijms23126708_

Round 1

Reviewer 1 Report

The manuscript entitled “Overexpression of Neurogenin 1 negatively regulates osteoclast and osteoblast differentiation” written by Kim et al., identified the significant role of Neurogenin 1 on bone remodeling, the experiments are well designed and controlled to prove the essential role of Ngn1 on bone resorbing osteoclast cells and bone forming osteoblast cells,

The manuscript was well written and interesting,

1)      The picture quality was very poor, should improve. The multinucleated osteoclast cells in fig1A and Fig4a, and cells stained for Alizarin Red were not clear,

2)      It would have been more appropriate if author used RAW264.7 cells for osteoclast and MC3T-E1 cells for osteoblast for transducing the overexpression of Ngn1 to analysis the transcription factor, it is easy to transduce the constructs in RAW & MC3T3 cells, I don’t know the rational behind using 293T cells, for luciferase assay, I could understand, but why for NFATc1, Runx2, Immunoprecipitation and Immunoblotting?

Author Response

Dear Reviewer,

We thank you for your thoughtful suggestions and insights. The manuscript has benefited from these insightful suggestions.

Reviewer 2 Report

Jung Ha Kim et al. demonstrated that overexpression of neurogenin 1 (Ngn1) in murine bone marrow-derived macrophages suppressed RANKL-induced osteoclastogenesis. Additionally, overexpression of Ngn1 in murine osteoblast precursors inhibited osteogenesis induced by osteogenic media. Furthermore, they showed that overexpression of Ngn1 suppressed TNF-a and IL-17A induced osteoclastogenesis and IL-17A-induced osteogenesis. They further demonstrated the inhibitory mechanism was associated with the binding of Ngn1 with PCAF (an acetyltransferase), which reduced the acetylation of NFATc1 and Runx2 (important transcriptional factors associated with osteoclastogenesis and osteogenesis). This is an interesting and well-designed paper. I have some recommendation to the authors.

Introduction:

1.     The authors mentioned Ibsp gene in figure 2c and mentioned OSE gene in figure 3e. These genes should be explained in the introduction with full name of Ibsp and OSE.

2.     They should describe the role of PCAF, and the role of acetylation in regulation of cell growth, osteoclastogenesis, osteogenesis, inflammation, and diseases.

Results

1.     Some figures (Fig1a, Fig4a, Fig5a, Fig5b) have low resolution and low magnification and it is hard to view the osteolclasts in these figures. I suggest to increase either the size of photos or increase the resolution of the pictures.

2.     The authors only showed ALP activities in Fig. 2a, Fig. 4b, Fig. 5c without showing the ALP staining in cells.  I suggest to include both ALP staining pictures and quantification of ALP activities in the results.

Discussion

The section is short. The authors should explain the following:

1.     What are the targets of PCAF acetylation?

2.     What are the other benefits of overexpression of Ngn1? What are the potential side effects of overexpression of Ngn1?

3.     Have other people conducted overexpression of Ngn1 in animals? What the results of overexpression of Ngn1 in other studies?

Author Response

(The authors gave the same response as above.)
